# Response of Crop Performance and Yield of Spring Sweet Potato (*Ipomoea batatas* [L.] Lam) as Affected by Mechanized Transplanting Properties

Hui Li [1], Baoqing Wang [2], Song Shi [1,2], Jilei Zhou [1,2], Yupeng Shi [1], Xuechuan Liu [1], Hu Liu [1,3,*] and Tengfei He [1,*]

[1] Shandong Academy of Agricultural Machinery Sciences, Jinan 250100, China; lihuictrc@shandong.cn (H.L.); shisong@shandong.cn (S.S.); zhoujilei@shandong.cn (J.Z.); shiyupeng1997@163.com (Y.S.); xuechuan_13@163.com (X.L.)

[2] Shandong Academy of Agricultural Sciences, Jinan 250100, China; xb970607@163.com

[3] Huang Huai Hai Key Laboratory of Modern Agricultural Equipment, Ministry of Agriculture and Rural Affairs, Jinan 250100, China

[*] Correspondence: liuhu0725@163.com (H.L.); hetengfei.1@163.com (T.H.); Tel.: +86-531-88617528 (H.L. & T.H.)

**Abstract:** The sweet potato transplanters of diverse transplanting configurations have been shown to produce various planting properties in relation to different raised bed cropping systems, thus affecting crop growth and yield in sweet potato cultivation. In Shandong Province, a field experiment assessed the effects of three treatments (RB1, mulched raised beds with a finger-clip type transplanter; RB2, bare raised beds with a finger-clip type transplanter; and RB3, bare raised beds with a clamping-plate type transplanter) on soil temperature, plant growth, yield, and economic benefits. With the lowest coefficient variation of plant spacing and planting depth, the RB1 with the finger-clip type transplanter had 6.4% and 6.0% higher temperature at 5–10 cm soil layer by using the plastic-mulch for rapid early slips growth as compared with the RB2 and the RB3, respectively. Consequently, the leaf area index in the RB1 was increased by 5.6% and 6.4% as compared to the RB2 and the RB3, separately. This finally contributed to 57.5–70.8% greater fresh vines weight and 23.8–33.8% higher tubers yield in the RB1 compared with both the RB2 and the RB3 treatments, respectively. In general, in the mulched raised bed system of the Huang-Huai-Hai region of China, the finger-clip type transplanter could be a suitable option for the transplanting of sweet potato slips. In the bare raised bed system, meanwhile, the clamping-plate type transplanter has the potential to increase the production of sweet potatoes.

**Keywords:** crop performances; planting properties of sweet potato transplanter; planting system; yield

## 1. Introduction

Food security is one of the greatest challenges facing humankind [1]. Agriculture is at the forefront of these challenges [2]. Sweet potato (*Ipomoea batatas* [L.] Lam) is one of the five most important crops in the world, rich in carbohydrates, and can serve as a source of protein, carotenoid, and essential vitamins for the survival needs of mankind [3,4]. This crop is widely cultivated from tropical to temperate regions, such as Asia, Africa, and Latin America [5,6]. There is an increasing need to produce more sweet potatoes on existing arable land given the challenges of both labor scarcity and population growth [7].

Sweet potato yields can vary significantly due to factors such as the soil, weather, crop variety, and cultivation management [8,9]. Under certain soil, weather, and sweet potato variety conditions, many efforts have been made to find cultivation modes that are more effective at enhancing productivity. Parwada et al. [10] established the proper ridging height and planting orientation in order to enhance constant reliable root yield and vine length among sweet potato producing farmers in Zimbabwe. Chagonda et al. [11] proposed

that the horizontal vine orientation provided a significant storage root diameter, while there was no significant difference between the ridge tillage and mound tillage systems. Abdallah et al. [12] evaluated the performance of sweet potato clones under different watering strategies in the coastal lowlands of Kenya. Ribeiro et al. [13] conducted a study to evaluate the plant growth, yield, uptake, and removal of N by sweet potato plants fertilized with N and treated with paclobutrazol during two planting seasons. Pepó [14] showed that a 0.75 m row spacing was more favourable than a 1.0 m one in Hungary.

China is the largest producer of sweet potatoes in the world [15]. Sweet potatoes are widely cultivated in over half of the globe's poor counties due to their wide ecological adaptation, strong tolerance to drought, and low requirement of soil fertilizer [16]. The cultivation areas for sweet potatoes in China are generally divided into the northern China area, the Yangtze River area, the southern China area, etc., which are distinguished by climatic conditions, cultivation systems, and soil conditions [17]. As shown in Figure 1, the Huang-Huai-Hai region of China is one of the most important traditional sweet potato production regions in China, accounting for 30% of national sweet potato production [18,19]. Many studies have shown that sweet potato cultivation on raised beds mulched with plastic film can be beneficial to sweet potato yield because it improves soil water moisture, soil bulk density, and soil porosity [20,21]. At present, farmers plant sweet potato on bare raised beds or raised beds mulched with plastic film in this area [17].

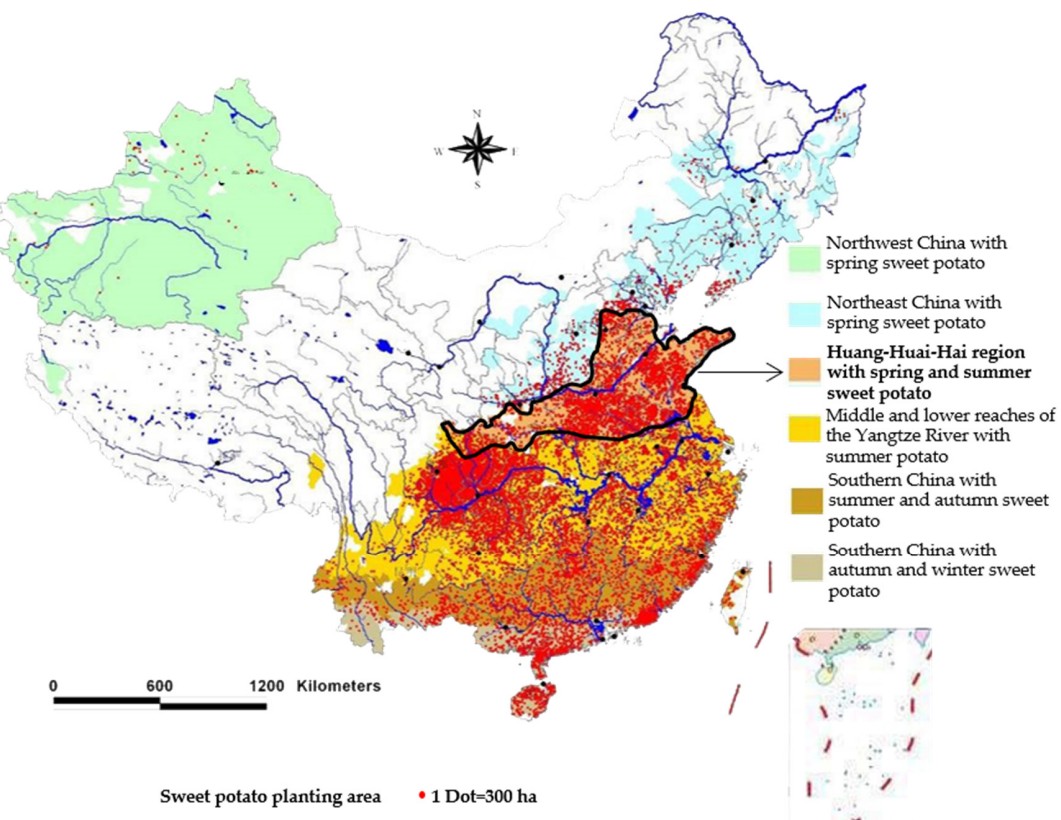

**Figure 1.** Traditional regional distribution and planting area of sweet potato cultivation in China.

However, most sweet potato production in the Huang-Huai-Hai region of China still occurs by the use of manual transplanting, which has caused this area to suffer from a labor shortage [22]. Sustainably producing the sweet potato crop in this region is thus a great challenge. There were, indeed, not even any special transplanters for transplanting sweet potato slips until Chen et al. [23] and Hu et al. [24] modified and improved the commercial clip-on-chain type transplanter for the horizontal transplanting of sweet potatoes in bare raised beds. These are mainly applicable for the bare raised bed cultivation system in

rain-fed farming areas that have high soil moisture and that are rainy, so no additional watering is required. The two machines cannot be used to mechanically transplant the sweet potato slips in drought-affected areas, though, due to the lack of a timely watering function. Supplementary irrigation is thus required at the time of planting for proper sprouting and establishment, although the prolific root system of sweet potato does make it a drought-tolerant crop [25,26]. Since the Huang-Huai-Hai region of China has limited water resources, there is a need for sweet potato transplanters in this area to accomplish the planting operation for raised beds mulched with plastic film system and the bare raised bed system. After several years of development, sweet potato transplanters with a slip taking-planting mechanism have been developed [27,28], and some have now been manufactured commercially. These transplanters have encouraged the development and extension of sweet potato production in Huang-Huai-Hai region of China, but the literature contains little information about their impact on planting properties and crop performance [29,30]. This paper compares two of the most widely used sweet potato transplanters (the finger-clip and the clamping-plate ones) for different planting modes (raised beds mulched with plastic film or the bare raised bed systems with varying placement), and it investigates their effects on planting quality, crop growth, and subsequent yield in 2021 in the Huang-Huai-Hai region of China.

## 2. Materials and Methods

### 2.1. Equipment Description

#### 2.1.1. Finger-Clip Compound Transplanter

The finger-clip compound transplanter, designed by the Shandong Academy of Agricultural Machinery Sciences and Shandong Huorong Agricultural Technology Development Co., Ltd. (Qingzhou, China), was used for sweet potato slips cultivation of bare raised beds and mulched raised bed systems. It mainly comprises a transmission box, a rotary component, a ridging board, a film pressing wheel, a height adjustment mechanism, a slip taking-planting mechanism, a drive system, a slip delivery mechanism, etc. (Figure 2a). It can accomplish land preparation, ridging, film mulching, drip-irrigation belt laying, and transplanting on two ridges at the same time. During transplanting, the rotary component completes the soil crushing and the soil preparation operations at 300~350 r/min, driven by power from the transmission box, which is connected to the tractor's power take off (PTO). The ridge board squeezes the crushed soil to form two rows of trapezoidal ridges with a height of 30 cm at 85 cm spacing under the traction of the tractor and the pressure of the hydraulic cylinder simultaneously. The drip irrigation laying device and the plastic-film frame mulches the ridge and lays the drip irrigation belt, respectively, and then the slip transplanting apparatus transplants the sweet potato slips by using the slip taking planting mechanism and the slip delivery mechanism at a rotary speed of less than 60 r/min, driven by the ground wheel. Slips are manually placed in the seedling delivery mechanism by the operators sitting on the seats.

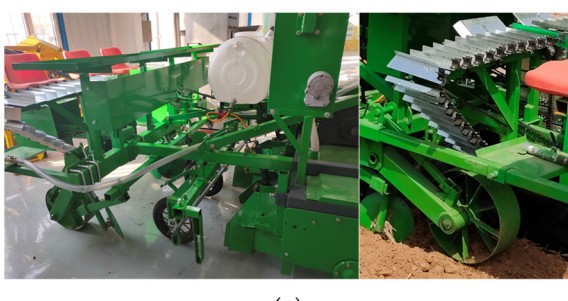

(**a**)

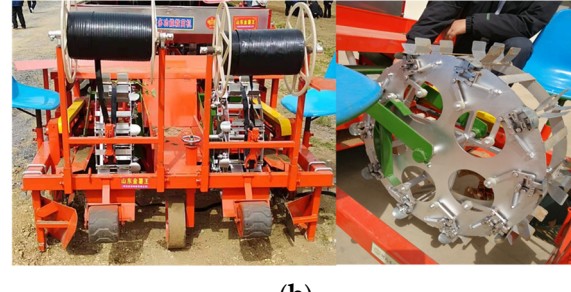

(**b**)

**Figure 2.** Two kinds of transplanters used for the experiment: (**a**) finger-clip compound transplanter with its slip transplanting apparatus; (**b**) clamping-plate compound transplanter with its seedling delivery mechanism.

### 2.1.2. Clamping-Plate Compound Transplanter

The clamping-plate compound transplanter (Shandong Jinshuwang Agricultural Machinery Manufacturing Co., Ltd., located in Tengzhou, China) is made up of a suspension frame, rotary blades, a ridge plough, a driving shaft, a slip conveying clamping-plate, a gear box, a soil loader, a slip fixing wheel, etc. (Figure 2b). The transplanter mounts with the tractor by the suspension frame. During the operation, the rotary blades smash soil at 340~360 r/min, driven by the tractor's PTO shaft. The soil is raised and enclosed by the ridge plough to form the raised beds of 30 cm height. Operators place the sweet potato slips in the seedling clips, which are installed on the slip conveying the clamping plate. The slips are then put horizontally vertical on the raised beds with the rotation of the conveying clamping-plate at 30~40 r/min. After this, the sweet potato slips remain covered with soil delivered by the soil loader. The fixing wheel presses the soil over the slips to finish the transplanting in the bare raised bed system. The key parameters of these two transplanters are presented below in Table 1.

**Table 1.** The key parameters of the two compound transplanters for sweet potato slips.

| Parameter | Finger-Clip Compound Transplanter | Clamping-Plate Compound Transplanter |
| --- | --- | --- |
| Matched power | 120–180 hp | 120–180 hp |
| Working width | 1.7 m | 1.7 m |
| Number of ridges | 2 | 2 |
| Transplanting part | Finger-clip type slip taking-planting mechanism | Clamping-plate type slip taking-placing mechanism |
| Transplant spacing | 20–30 cm | 20–30 cm |
| Transplanting depth | 4–10 cm | 4–10 cm |
| Slips placement | Boat-shape placement | Horizontal vertical placement |
| Suitable system | Mulched raised beds system and bare raised beds system | Bare raised beds system |
| Productivity | 0.08–0.13 ha h$^{-1}$ | 0.1–0.2 ha h$^{-1}$ |

### 2.2. Site Description

Field trials were conducted at Zhangqiu (36°41′ N, 117°32′ E), located in the southeast of the Huang-Huai-Hai region of China, with three crop rotation treatments. In the five years before the experiment, this area had a monsoon climate with an annual average temperature of 10~20 °C, a frost-free period of 167~218 days, and annual rainfall of 450~1100 mm. The accumulated temperature of ≥0 °C is about 5401 °C [31]. In this double cropping area, winter wheat to summer maize is the main crop rotation. When the sweet potato was planted, the winter wheat (end of September to the middle of June) to summer maize (middle of June to end of September) to spring sweet potato (end of April or early May to end of September) rotation is used. According to the USDA texture classification system, the soil in the experiment plots is silt loam, clay (12.3%), silt (74.8%), and sand (12.9%), on average. In the top 30 cm soil layer, soil bulk density, soil moisture, and pH were 1.35 g/cm$^3$, 12.8%, and 8.3, respectively.

### 2.3. Experimental Design

In the experiment, three treatments were compared: the finger-clip compound sweet potato transplanter for the mulched raised beds system (RB1) (Figure 3a), the finger-clip compound sweet potato transplanter for the bare raised beds system (RB2) (Figure 3b), and the clamping-plate compound sweet potato transplanter for the bare raised bed system (RB3) (Figure 3c). The three treatments were designed in a randomized block with 3 replications. Each plot was 3.5 m wide and 30 m long with an access pathway and guard strip between each. The spring sweet potato slips (variety Jishu 26, and a length of 30 cm~35 cm) with five top nodes were transplanted on 6–7 May and harvested on 8–9 October. Drip irrigation was immediately applied after the transplanting. In the RB1 system, a high-density black polyethylene film (0.02 mm thick, 1.0 m wide) was used as the mulching plastic. In the treatments RB1 and RB2, the sweet potato slips were transplanted as a boat-shape along

the ridge direction by using the finger-clip compound sweet potato transplanter, while the sweet potato slips were transplanted as a horizontal vertical placement in the treatment RB3 by using the clamping-plate compound sweet potato transplanter.

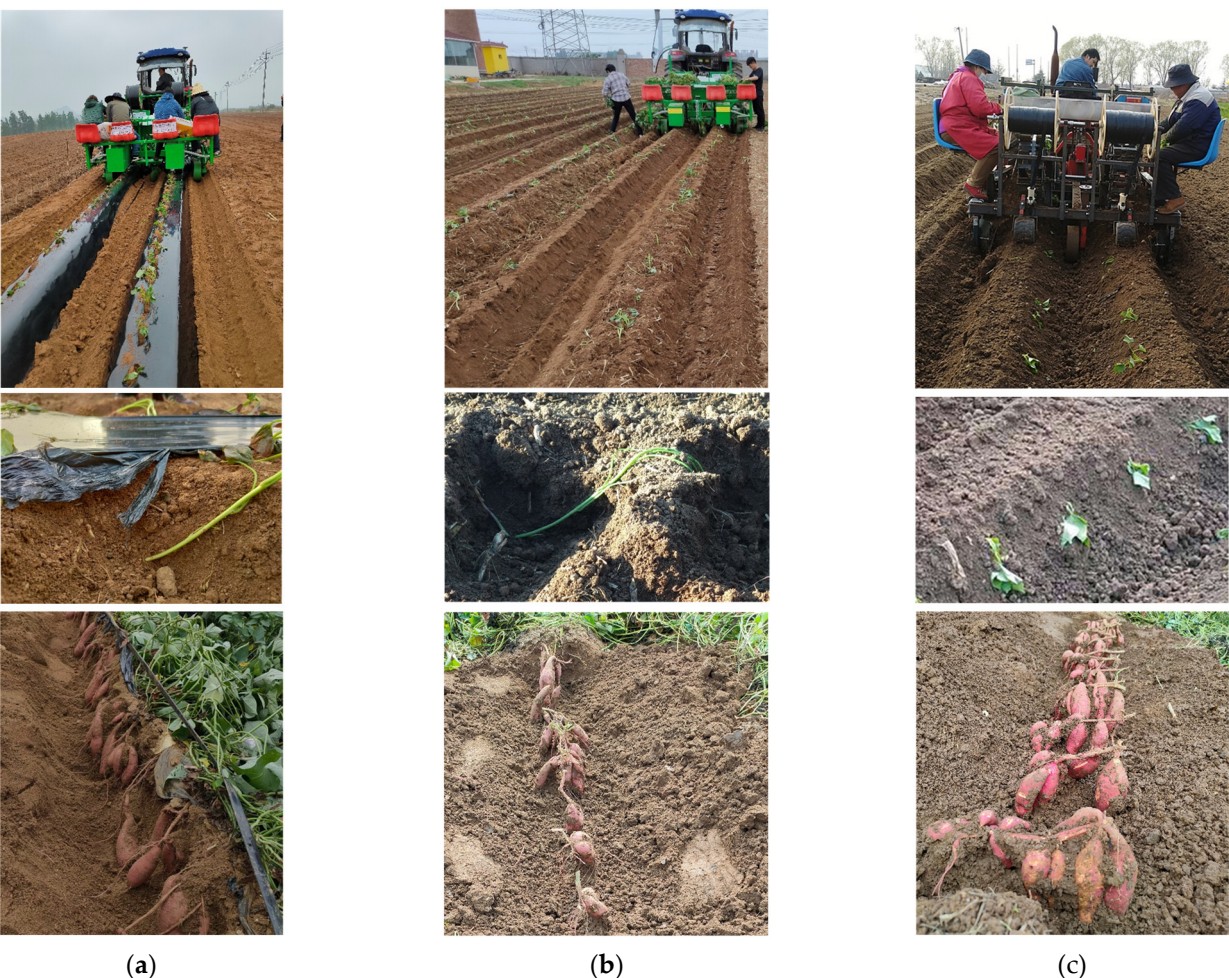

| (a) | (b) | (c) |

**Figure 3.** Three designed treatments in this experiment: (**a**) RB1, finger-clip compound transplanter working under mulched raised beds system; (**b**) RB2, finger-clip compound transplanter working under bare raised beds system; (**c**) RB3, clamping-plate compound transplanter working under bare raised beds system.

The sweet potato slips were planted with the district-recommended plant density of about 49,000 plants/ha with 24 cm × 85 cm plant spacing and planting depth of 5~10 cm. The compound fertilizer [N-P$_2$O$_5$-K$_2$O 10-8-24] (containing total nutrients ≥ 42%, humic acid ≥ 3%, controlled-release K fertilize ≥ 4%) was applied as the basal fertilizer at the rate of 375 kg/ha at transplanting, while 33% pendimethalin EC herbicide (JiangSu Longdeng Chemical Company, Kunshan, China) was sprayed onto the soil surface according to the manufacturer's protocol during the transplanting. About 1.5 months after the transplanting, 80% flumetsulam WG herbicide (Jiangsu Ruibang Pesticide Factory Co., Ltd., Changzhou, China) was carefully used in the three treatments.

*2.4. Measurements*

2.4.1. Missing Seedling Rate and Qualified Rate of Transplanting Population

The missing seedling rate and the qualified rate of the transplanting population, representing the transplanting quality, were counted—-120 theoretical sweet potato slips

that should be planted at the 12 split-plots in 3 complete randomized blocks [32,33]. They were calculated using the following equations:

$$Q_M = \frac{N_{LZ}}{N'} \times 100\% \qquad (1)$$

$$Q_z = \frac{N_T - (N_{LM} + N_{MM} + N_{CZ} + N_{SM})}{N'} \times 100\% \qquad (2)$$

where $Q_M$ is the missing seedling rate (%), $Q_Z$ is the qualified rate of the transplanting (%), $N_T$ is the total planted counts of the sweet potato slips, $N'$ is the theoretically planted counts, $N_{LZ}$ is the missed planted counts of the sweet potato slips, $N_{LM}$ is the exposed planted counts of the slips, $N_{MM}$ is the buried counts of the planted slips, $N_{CZ}$ is the replanted counts of the slips, and $N_{SM}$ is the injured counts of the planted slips.

### 2.4.2. Precision of Seedling Placement

To calculate the plant spacing of the sweet potato slips, 60 successively planted sweet potato slips were measured of the randomly selected planting row in each plot. To calculate the seeding or the planting depth of crops, the chlorophyll-free stem and coleoptile length (from seed remnants to the onset of green stem) was usually measured as effective depth [34]. For sweet potato slips, the chlorophyll-free stem lengths were not obvious. After 10 days of planting, a mark was made on the five seedlings at the ridge level in each plot. The vertical distance from the lowest position to the marked point was taken as the effective planting depth, and then the sweet potato slips were dug out and the entire stem length below the mark was taken as the effective planting length. The mean planting length was easily obtained. The plant spacing coefficient of variation and the qualified rate of transplanting depth were calculated to assess the transplanting accuracy in each plot using the following equations [32]:

$$CV_X = \frac{\sqrt{\frac{1}{n-1} \sum_{i=1}^{n} \left(X_i - \frac{\sum_{i=1}^{n} X_i}{n}\right)^2}}{\frac{\sum_{i=1}^{n} X_i}{n}} \times 100\% \qquad (3)$$

$$V_H = \frac{N_h}{N_T} \times 100\% \qquad (4)$$

$$CV_H = \frac{\sqrt{\frac{1}{n-1} \sum_{i=1}^{n} \left(H_i - \frac{\sum_{i=1}^{n} H_i}{n}\right)^2}}{\frac{\sum_{i=1}^{n} H_i}{n}} \times 100\% \qquad (5)$$

where $CV_X$ is the plant spacing coefficient of variation; $n$ is the measured number of the planted slips; $X_i$ is the measured plant spacing, cm; $V_H$ is the qualified rate of transplanting depth; $N_h$ is the sweet potato counts of qualified depth; $CV_H$ is the plant depth coefficient of variation; and $H_i$ is the measured plant depth, cm. As the designed transplanting depth was 60 mm, we assumed that the qualified depth was $60 \pm 10$ mm.

### 2.4.3. Soil Temperature and Plant Growth

In different treatments, soil temperature was measured at 5 and 10 cm soil depths at 08:00 ($T_{8:00}$), 14:00 ($T_{14:00}$), and 20:00 ($T_{20:00}$). A high precision soil temperature and humidity sensor (JXBS-3001-TR), connected with the weather station, was used. The mean daily soil temperature ($T$) for 10 days during the period from 10 days to 1 month after transplanting was calculated as follows [35]:

$$T = (2 \times T_{8:00} + T_{14:00} + T_{20:00})/4 \qquad (6)$$

The leaf number, the plant height, and the leaf area were all measured to estimate the growth of the spring sweet potato. The samples were measured and obtained within randomly selected areas of 1 m × 1 m from three areas in each plot 1 month after planting. Plant height was calculated from the stem tip to the soil surface. To obtain the leaf area, the leaves were cut and analysed by the LA-S series plant image analysis system (Hangzhou Wanshen Testing Technology Co., Ltd., Hangzhou, China) in a laboratory. After that, the Leaf Area Index (LAI) was calculated as follows [36]:

$$LAI = LA/GA \tag{7}$$

where LAI is the leaf area index, LA is the leaf area in the selected area ($m^2$), and GA is the ground area of the selected area ($m^2$).

### 2.4.4. Weight of Fresh Vines with Leaves and Tuber Yield

At harvest time (i.e., the beginning of October), the weight of the vines with leaves, the number of vines, and the length of the longest vine per plant, which were removed manually in the experiment, were all measured [37]. In each plot, we chose 10 plants randomly.

The tuber yield that was observed in this study included the number of tubers (per plant), the fresh weight of tuber (g $plant^{-1}$), and the yield (t $ha^{-1}$). During manual harvesting, we collected 10 plants, with an area of the harvest bed that was 170 cm wide and 120 cm long (sampling size), which was taken randomly from each plot. The average number of tubers per plant was measured and categorized as large marketable tubers ($\geq$500 g), medium marketable tubers ($\geq$200 g), and non-marketable tubers (<200 g, or else damaged by insects and diseased tubers) [16]. The total yield per hectare was then calculated using the following equation [38,39]:

$$\text{Yield (t } ha^{-1}) = (10{,}000/\text{scale of sampling plot}) \times \text{yield of sampling plot} \tag{8}$$

### 2.4.5. Economic Benefit

Input (sweet potato slips, fertiliser, labour, etc.) quantities and the direct cost of all mechanical operations was recorded throughout the field trial, together with the value of outputs (crop yield value), on a common basis (US$ $ha^{-1}$) [40].

### 2.5. Data Analysis

The SPSS analytical software package was used for all of the statistical analyses. Mean values were calculated for each of the measurements, and ANOVA was used to assess the effects of the two sweet potato transplanters on both the planting properties and the crop performance of the measures. When the ANOVA indicated a significant F-value, multiple comparisons of annual mean values were performed by the least significant difference (LSD) method. In all analyses, a probability of error smaller than 5% ($p = 0.05$) was considered statistically significant.

## 3. Results

### 3.1. Missing Seedling Rate and Qualified Rate of Transplanting Population

Table 2 shows that the mean missing seedling rate $Q_M$ under RB3 treatment of 0.6% appeared to be 59.7% and 77.6% lower ($p > 0.05$) than that under RB1 treatment of 1.4% and RB2 treatment of 2.5%, respectively. This difference was only relevant to the missed counts of the sweet potato slips, while the theoretical planted counts were the same according to Formula (1). To evaluate transplanting quality, the replanted count number $N_{CZ}$ in RB3 of 0.7 was significantly ($p < 0.05$) greater than that in both RB1 and RB2 treatments. However, the difference of exposed counts $N_{LM}$, buried counts $N_{MM}$, and injured counts $N_{SM}$ of the planted sweet potato slips were all non-significant ($p = 0.05$) under the three treatments. The qualified rates were also similar in the three treatments.

**Table 2.** The transplanting quality under the three treatments. Means within a column followed by the same letters are not significantly different ($p = 0.05$).

| Treatment | Mean Value | | | | | | | Transplanting Quality | |
|---|---|---|---|---|---|---|---|---|---|
| | Theoretical Planted Counts $N'$ | Total Planted Counts $N_T$ | Missed Counts $N_{LZ}$ | Exposed Counts $N_{LM}$ | Buried Counts $N_{MM}$ | Replanted Counts $N_{CZ}$ | Injured Counts $N_{SM}$ | Missing Seedling Rate $Q_M$ (%) | Qualified Rate $Q_Z$ (%) |
| RB1 | 120.0 a | 118.3 a | 1.7 a | 2.7 a | 0 a | 0 a | 0.3 a | 1.4 a | 96.1 a |
| RB2 | 120.0 a | 117.0 a | 3.0 a | 2.0 a | 0.3 a | 0 a | 0 a | 2.5 a | 95.6 a |
| RB3 | 120.0 a | 119.7 a | 0.7 a | 3.3 a | 0 a | 0.7 b | 0 a | 0.6 a | 96.1 a |

*3.2. Precision of Seedling Placement*

The planting spacing in the RB1 and RB3 treatments were marginally higher ($p > 0.05$) than that in the RB2 treatment (Table 3). However, the plant spacing coefficient of variation in RB1 of 5.1% was 75.2%, significantly smaller ($p < 0.05$) than that in RB3 treatment of 8.9%. The mean planting length in the three treatments were all around 200 mm. The planting depth in the RB1 treatment was 4.0% ($p > 0.05$) and 32.2% ($p < 0.05$) deeper than that in the RB2 and RB3, respectively, and the relative coefficient of variation was slightly lower than in the other treatments. Meanwhile, the qualified rate of the planting depth in the three treatments was nearly the same, and all were above 95%.

**Table 3.** Precision of seedling placement under the three treatments. Means within a column followed by the same letters are not significantly different ($p = 0.05$).

| Treatment | Plant Spacing | | Mean Planting Length (mm) | Planting Depth | | |
|---|---|---|---|---|---|---|
| | Spacing Value (cm) | Coefficient of Variation (%) | | Depth Value (mm) | Qualified Rate (%) | Coefficient of Variation (%) |
| RB1 | 24.3 a | 5.1 a | 201.8 a | 78.1 a | 97.1 a | 8.7 a |
| RB2 | 23.9 a | 6.1 ab | 198.2 a | 75.4 a | 96.9 a | 9.2 a |
| RB3 | 24.2 a | 8.9 b | 202.9 a | 59.2 b | 97.5 a | 10.6 a |

*3.3. Soil Temperature and Plant Growth*

In general, a soil temperature at 5 cm depth was marginally higher than that at 10 cm depth in the three treatments (Table 4). At 5 cm depth, the RB1 increased soil temperature by 0.3–1.5 °C and 0.1–1.5 °C, respectively, as compared to the RB2 and the RB3 treatments within the month after the transplanting day. The soil temperature was 5.2%, significantly higher in the RB1 than in the RB2 treatment on the 30th day after transplanting. The difference between the RB1 and the RB3 on the 10th day and the 30th day was significant at $p = 0.05$ level, independently. Similar results were found in the 10 cm soil depth where RB1 increased the temperature by 6.4% and 6.0% as compared to the RB2 and the RB3, respectively, on the 30th day after transplanting.

As shown in Table 5, the difference of leaf number, plant height, and leaf area were all not significant ($p > 0.05$) in the RB1, RB2, and RB3 treatments 1 month after transplanting. The leaf number in the RB1 was 10.8% and 26.3% higher than that in the RB2 and the RB3, relatively, and the RB1 increased the plant height by 8.7% and 6.4% as compared with the RB2 and the RB3, respectively. Meanwhile, the leaf area index in the RB1 was increased by 5.6% and 6.4% compared to the RB2 and RB3 treatments.

**Table 4.** Soil temperature at 5 cm and 10 cm depth soil layer in three treatments. Means within same transplanting days in the same soil layer followed by the same letter are not significantly different (*p* = 0.05).

| Treatment | Soil Layer Depth (cm) | Mean Daily Soil Temperatures (°C) | | |
|---|---|---|---|---|
| | | 10 Days after Transplanting | 20 Days after Transplanting | 30 Days after Transplanting |
| RB1 | | 20.0 a | 23.4 ab | 29.4 b |
| RB2 | 5 | 19.0 a | 23.1 a | 28.0 a |
| RB3 | | 18.6 a | 23.3 b | 28.0 b |
| RB1 | | 18.3 a | 21.5 a | 27.5 a |
| RB2 | 10 | 17.2 a | 21.1 a | 25.8 a |
| RB3 | | 17.2 a | 21.4 b | 25.9 b |

**Table 5.** Plant growth of the three treatments one month after transplanting. Means within a column followed by the same letters are not significantly different (*p* = 0.05).

| Treatment | Leaf Number | Plant Height (mm) | Leaf Area Index |
|---|---|---|---|
| RB1 | 7.2 a | 83.6 a | 0.125 a |
| RB2 | 6.5 a | 76.9 a | 0.118 a |
| RB3 | 5.7 a | 78.6 a | 0.117 a |

*3.4. Weight of Fresh Vines with Leaves and Tuber Yield*

As shown in Table 6, the RB1 treatment had 7.7% (*p* > 0.05) and 30.2% (*p* < 0.05) more branches in the growth period of nearly five months as compared with the RB2 and RB3, respectively. Meanwhile, the relative weight of the fresh vines with leaves in the RB1 was significantly (*p* < 0.05) increased by 57.5% and 70.8% compared to that in the RB2 and the RB3, respectively. However, the length of the longest vine of each plant was similar (1.5–1.7 m), which may be determined by the growth characteristics of the same sweet potato variety.

**Table 6.** Weight of fresh vines and tuber yield in three treatments during the experiment. Means within a column by the same letters are not significantly different (*p* = 0.05).

| Treatment | Vines (/Plant) | | | Tubers (/Plant) | | | | | Yield (t ha$^{-1}$) |
|---|---|---|---|---|---|---|---|---|---|
| | Total Number | Length of Longest Vine (m) | Weight of Fresh Vines (g) | Total Number | Large Tubers No. | Medium Tubers No. | Fresh Weight (g) | Standard Deviation | |
| RB1 | 5.6 a | 1.7 a | 949.7 a | 4.0 a | 1.0 a | 3.0 a | 875.2 a | 27.0% | 42.9 a |
| RB2 | 5.2 ab | 1.5 a | 602.8 b | 4.2 a | 1.0 a | 3.0 a | 653.8 a | 24.5% | 32.1 a |
| RB3 | 4.3 b | 1.7 a | 556.1 b | 5.2 a | 0.9 a | 4.2 a | 706.8 a | 26.6% | 34.6 a |

In this research, the mean tuber number per plant in each treatment was 4–5, while the number of large tubers was about 1 and the number of nedium tubers was about 3–4. The weight of single tubers in the RB3 was slightly more uniform than that in the RB1 treatment, even when it had higher variation of plant spacing and planting depth. The tuber yield per plant was 875.2 g plant$^{-1}$ in the RB1 compared to 653.8 g plant$^{-1}$ in the RB2 and 706.8 g plant$^{-1}$ in the RB3, which indicated that the tubers yield in RB1 was 23.8–33.8% higher than that in the RB2 and the RB3. As a result, the fresh tuber yield was 32.1–42.9 t ha$^{-1}$ in the three treatments.

*3.5. Economic Benefit*

As shown in Table 7, mean annual input costs for the three treatments varied from 3203.0 US\$ ha$^{-1}$ in RB2 to 3337.6 US\$ ha$^{-1}$ in the RB1. The RB1 cost the most due to using plastic mulch, even though it used less herbicide and water. Meanwhile, the RB3 cost

the least in terms of the labour use of transplanting due to the higher productivity of the machine. However, the difference of input costs among the three treatments was marginal. Since the RB1 had a greater fresh tuber yield, the farmer profit for the RB1 was 43.8% and 30.2% greater than that for the RB2 and the RB3, respectively.

**Table 7.** Economic benefit analysis for three treatments.

| Treatment | RB1 | RB2 | RB3 |
|---|---|---|---|
| Inputs | | | |
| Sweet potato slips (US$ ha$^{-1}$) | 765.6 | 765.6 | 765.6 |
| Fertilizer (US$ ha$^{-1}$) | 210.9 | 210.9 | 210.9 |
| Herbicide (US$ ha$^{-1}$) | 81.8 | 93.8 | 93.8 |
| Plastic mulch and drip irrigation pipe (US$ ha$^{-1}$) | 632.7 | 485.1 | 485.1 |
| Mechanical operation cost in transplanting (US$ ha$^{-1}$) | 234.4 | 234.4 | 234.4 |
| Labour in transplanting (US$ ha$^{-1}$) | 125.0 | 117.2 | 113.3 |
| Irrigation (US$ ha$^{-1}$) | 21.6 | 30.4 | 37.8 |
| Mechanical operation cost in other process (US$ ha$^{-1}$) | 703.1 | 703.1 | 703.1 |
| Labour use in other process (US$ ha$^{-1}$) | 562.5 | 562.5 | 562.5 |
| Total (US$ ha$^{-1}$) | 3337.6 | 3203.0 | 3206.5 |
| Outputs | | | |
| Yield (US$ ha$^{-1}$) | 42.9 | 32.1 | 34.6 |
| Price (US$ kg$^{-1}$) | 0.39 | 0.39 | 0.39 |
| Income (US$ ha$^{-1}$) | 16,731.0 | 12,519.0 | 13,494.0 |
| Farmer income (US$ ha$^{-1}$) | 13,393.4 | 9316.0 | 10,287.5 |

## 4. Discussion

The clamping-plate compound sweet potato transplanter had the least missed transplanting counts and the greatest exposed transplanting counts (Table 2). This was due to the reduced action of taking-planting the sweet potato slips, which was one of the typical differences between the clamping-plate type and finger-clip type compound sweet potato transplanters. In the RB3, the planting depth (59.2 mm) was the shallowest and its variation was the greatest, as shown in Table 3. The reason for this is that sweet potato slips were placed on the ridge through lifting and through covering the soil on the slips by using the clamping-plate compound sweet potato transplanter. The plant spacing variation (8.9%) of the clamping-plate compound sweet potato transplanter by using the soil-covering method was in agreement with that of the other horizontal transplanter [41], which has a similar method of placing the slips. The transplanting depth qualified rate (96.9–97.1%) and planting length (198.2–201.8 mm) in the RB1 and the RB2 of the finger-clip compound sweet potato transplanter were in accordance with Murakami et al. [27]. Available water for the plant is necessary for rapid early slips growth [10]. The shallower the slips were planted in the RB3 treatment, the more irrigation was needed. All the transplanting quality and precision in the RB1, RB2, and RB3 treatments satisfied the sweet potato transplanting requirements [17].

Soil temperature is an important environmental factor for plant growth and development [42,43]. The Huang-Huai-Hai region of China was usually suffering a sudden temperature drop from the end of April to the beginning of May. The soil temperature in the 5–10 cm soil layer of the RB1 treatment was 0.3–1.7 °C and 0.1–1.7 °C higher during the first month after transplanting than that of the RB2 and the RB3, respectively, with the help of the plastic mulch. Rao et al. [44] also pointed out that mean soil temperatures (19.9 °C) were significantly higher under mulched plots compared to non-mulched soil (19 °C) during their three-year experiment.

The proper soil temperature tended to promote the sweet potato growing processes, as shown previously by Bandara et al. [45]. The higher temperature in the RB1 treatment in the first transplanting month could help to produce better growing conditions, and the plant height and leaf area index were both improved in the RB1 treatment in the initial growing period in this study. The improvements may also be caused by the fact that more

moisture was retained and by enhanced mineral N (29–87%) in the mulched soil for the dry season, as previously indicated by Kundu et al. [46].

Mulched soil enhanced mineral N, P, and K availability is applied for sweet potato [46], while all of those chemical properties are critical for the yield increasing. The mulched raised beds in the RB1 contributed to the significant ($p < 0.05$) increase of 346.9 g plant$^{-1}$ of the aboveground growth and the marginal increase of 221.4 g plant$^{-1}$ of the fresh tubers weight compared with those of the RB2. These results are consistent with those found by Rao et al. [44]. Moreover, using plastic mulch in cool climates seems to increase the aboveground growth of sweet potato significantly, while the storage root yield was less affected [47]. In the RB3, the total tubers number was higher than the other treatments under the horizontal vertical placement by using the clamping-plate compound sweet potato transplanter. The increased number of the fresh tubers in the RB3 was offset by the decrease in the weight of each fresh tuber, and the size of the tubers was more consistent and more popular for fresh sweet potatoes. The yield in the RB3 was slightly higher than that of the RB2 while its weight of the fresh vines of each plant was marginally lower than the RB2, possibly due to the horizontal vertical placement with the varying slips orientation. In RB3, the slips were grown above the ridge furrow by being placed horizontally and vertically to the ridge, while in the RB2, the slips were grown above the ridge by being placed along the ridge during the first few growing months of the growing period. As a result, the distribution of the solar energy in the RB3 was much greater on the ridge areas than that of the RB2, which is crucial for tuber growth as they are planted in the ridge.

The positive effects of mulching and horizontal vertical placement on crop growth and yield were probably responsible for the increased economic benefits in the RB1 and the RB3 treatments. The results agree with those of Hou et al. [21] and Rao et al. [44]. The proportion of labor costs in the mechanized sweet potato production process of this study was 20.1%, which dropped significantly compared with the study of Kassali, in which no machine was used, and in which the labor cost accounted for 68% of the total cost [48]. The use of mechanization in sweet potato production increased the economic benefits. Tang et al. [49] also found that the labor cost was 46.5% of the total cost during sweet potato production in which the transplanting process was accomplished manually. It seems that the use of the mechanized transplanting reduced the labor cost by 26.5%. Yan et al. also pointed out that the labor volume of sweet potato transplanting accounts for about 23% of the whole production process [41], but the labor cost only accounted for 3.7% in this study because of the use of mechanical transplantation. The replacement of labor transplanting with mechanized transplanting thus contributed significantly to the improvement of the economics of sweet potato production.

## 5. Conclusions

In this study, considerable changes in crop performances and yield due to mulched raised beds and horizontal vertical transplanting placement were observed. The finger-clip compound sweet potato transplanter and the clamping-plate compound sweet potato transplanter satisfies the requirement of sweet potato transplanting among three raised bed cropping systems. With the lowest coefficient variation of plant spacing and planting depth, the finger-clip compound sweet potato transplanter produced the raised beds with a higher temperature by using the plastic mulch for the growth of rapid early slips in the RB1 treatment, thereby improving 57.5–70.8% of the weight of fresh vines and 23.8–33.8% of the yield of tubers compared to both the RB2 and RB3 treatments. However, when the plastic mulch was not used, the clamping-plate compound sweet potato transplanter provided a 7.8% higher yield than the finger-clip compound sweet potato transplanter by placing the slips horizontally vertical to the raised beds. In general, in the areas of the mulched soil planting system, the finger-clip compound sweet potato transplanter could be a suitable option. In the areas without mulch, though, the clamping-plate compound sweet potato transplanter has the potential to increase production.

**Author Contributions:** Conceptualization, H.L. (Hui Li) and B.W.; methodology, T.H.; machine preparation, J.Z. and H.L. (Hu Liu); field experiment, S.S., Y.S. and X.L.; validation, T.H.; data curation, H.L. (Hu Liu); writing—original draft preparation, H.L. (Hui Li); writing—review and editing, B.W.; project administration, H.L. (Hui Li); funding acquisition, H.L. (Hui Li). All authors have read and agreed to the published version of the manuscript.

**Funding:** This research was funded by the National Natural Science Foundation of China (Grant No. 32201683) and Agricultural Science and Technology Innovation Project of Shandong Academy of Agricultural Sciences (Grant No. CXGC2023B04).

**Data Availability Statement:** Not applicable.

**Acknowledgments:** Thanks to all of our workmates and the postgraduate students working in the Huang Huai Hai Key laboratory of Modern Agricultural Equipment, who provided their input on this research.

**Conflicts of Interest:** The authors declare no conflict of interest.

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
