# Peer review of "Response of Crop Performance and Yield of Spring Sweet Potato (Ipomoea batatas [L.] Lam) as Affected by Mechanized Transplanting Properties"

_agronomy, doi:10.3390/agronomy13061611_

Round 1

Reviewer 1 Report

·         I feel the title is too broad and confusing

·         Use past tense for things that you did, present for general theories and applications

·         Check the manuscript format to follow the Journal properties

·         In Figures, the split letters (A, B) come before the description

·         L251. “which is only relevant to the missed counts of the sweet potato slips.” ??? Explain

·         L272. - Write out numbers when less than 10 and not referring to units, e.g. three not 3

·         L273. Correct symbol for degrees. Add space between the number and the unit.

·         L300. Replace t/ha with t ha-1

·         L305. g plant-1 instead of g/plant.

·         Similar to L300 and L305 comment, correct the unit format of Table 6 and in the rest manuscript.

Author Response

Dear reviewer:

Thanks for your professional review. We polished our work according to your suggestions. And more details can be seen in the attachment.

Best regards! 

Hui Li

Reviewer 2 Report

This study presents the results of a field experiment which assessed the effects of three treatments (RB1, mulched raised beds with finger-clip type transplanter; RB2, bare raised beds with finger-clip type transplanter; RB3, bare raised beds with clamping-plate type transplanter) on soil temperature, plant growth, yield and economic benefits In Shandong province in China.

This is an interesting study with high significance of content, clearly presented results that support study's objectives. Only minor editing of English language is required. I recommend publication of this study once the issues noted in the attached manuscript will be taken care of.

Author Response

Dear reviewer,

First, thanks for the professional and careful review. The suggestions will be the best way to perfect our work. And we made the changes according to them. More details can be seen in the attachment.

Best regards!

Hui Li
